# Characterising Kenyan hospitals' suitability for medical officer internship training: a secondary data analysis of a cross-sectional study

Yingxi Zhao ![ORCID],[1] Boniface Osano,[2,3] Fred Were,[2] Helen Kiarie,[4]
Catia Nicodemo ![ORCID],[5] David Gathara,[6] Mike English ![ORCID] [1,7]

For numbered affiliations see end of article.

**Correspondence to**
Yingxi Zhao;
yingxi.zhao@ndm.ox.ac.uk

## ABSTRACT

**Objective** To characterise the capacity of Kenya internship hospitals to understand whether they are suitable to provide internship training for medical doctors.

**Design** A secondary data analysis of a cross-sectional health facility assessment (Kenya Harmonized Health Facility Assessment (KHFA) 2018).

**Setting and population** We analysed 61 out of all 74 Kenyan hospitals that provide internship training for medical doctors.

**Outcome measures** Comparing against the minimum requirement outlined in the national guidelines for medical officer interns, we filtered and identified 166 indicators from the KHFA survey questionnaire and grouped them into 12 domains. An overall capacity index was calculated as the mean of 12 domain-specific scores for each facility.

**Results** The average overall capacity index is 69% (95% CI 66% to 72%) for all internship training centres. Hospitals have moderate capacity (over 60%) for most of the general domains, although there is huge variation between hospitals and only 29 out of 61 hospitals have five or more specialists assigned, employed, seconded or part-time— as required by the national guideline. Quality and safety score was low across all hospitals with an average score of 40%. As for major specialties, all hospitals have good capacity for surgery and obstetrics-gynaecology, while mental health was poorest in comparison. Level 5 and 6 facilities (provincial and national hospitals) have higher capacity scores in all domains when compared with level 4 hospitals (equivalent to district hospitals).

**Conclusion** Major gaps exist in staffing, equipment and service availability of Kenya internship hospitals. Level 4 hospitals (equivalent to district hospitals) are more likely to have a lower capacity index, leading to low quality of care, and should be reviewed and improved to provide appropriate and well-resourced training for interns and to use appropriate resources to avoid improvising.

## Strengths and limitations of this study

⇒ Using data collected from a national health facility assessment, our study is able to characterise the capacity of 61 out of 74 Kenya internship hospitals and contrast findings across levels of hospitals and with the minimum requirements outlined in the national guidelines for medical officer internship training.

⇒ Our analyses have clear implications for Kenya policy-makers to improve the conditions of internship hospitals.

⇒ More widely, our findings point to the need to carefully consider the potential consequences of rapidly expanding medical training and of appropriate planning and financing for new internship centres, especially in rural areas.

⇒ Our analysis was only limited to the 61 internship training hospitals sampled by Kenya Harmonized Health Facility Assessment (KHFA), mostly public hospitals; and KHFA data were collected in 2018 prior to COVID-19 therefore it is possible that the capacity of Kenyan internship training centres has improved since 2018.

⇒ To assess the capacity we selected 166 signal indicators from the 3000 questions from KHFA survey, however the indicators selected focused only on the structural and organisational features of internship training hospitals, and the selection process could be somewhat subjective.

## BACKGROUND

In Kenya as in many countries, doctors are first trained for 5 or 6 years in medical school (typically 2 or 3 years preclinical, 3 years clinical). This is followed by a mandatory 1 year internship prior to receiving a license to practice from the national regulatory board, that is, the Kenya Medical Practitioners and Dentists Council, overseen by the National Ministry of Health.[1] The Kenyan medical internship included supervised rotation in four major departments (surgery, internal medicine, paediatrics and child health and obstetrics and gynaecology (OBGYN)) in one Internship training centre but since 2020 mental health and community health practice became added requirements. The competencies medical interns are expected to develop, for example, compulsory procedures that should be carried out either independently

or under supervision, are outlined in national guidelines and in greater detail in interns' personal log books.[2] After successfully completing internship and subsequent licensure, individuals can practice medicine unsupervised as general medical officers in different types of facilities.

Medical internships are undertaken in specific hospitals approved for this purpose after evaluation by the Kenya Medical Practitioners and Dentists Council, and all hospitals should meet and maintain their minimum requirements. These sites are an essential component of the health and education system producing a country's future physicians.[3] Ensuring internship training hospitals are well-staffed and equipped is important so that medical graduates can be supervised and supported to consolidate their knowledge and skills, including of common clinical procedures, and become competent medical doctors.[4] Improving internship training could also improve the quality of care and patient safety as interns are at the frontline of patient management especially in low-income and middle-income countries (LMICs).[5]

As with other African countries, medical schools in Kenya are expanding to meet human resources for health gaps. The number of Kenyan medical schools has increased dramatically from 2 in 2004 to 12 in 2018,[6] meanwhile the number of medical doctors graduating almost doubled from 287 in 2006 to 573 in 2018.[7 8] The number of internship training centres recognised by the Ministry of Health and regulators also increased from 6 to 74 over the past 15 years and includes private or mission/faith-based hospitals.[9 10] Many new training centres that were formerly district hospitals but are now county hospitals in Kenya (at level 4 of its six-level health system, usually primary facilities including district hospitals) became recognised as internship centres alongside more established secondary or tertiary regional and national hospitals (at level 5 and 6, respectively). However, these county hospitals may have shortages of staff, limited diagnostic facilities and frequent stockout of medications and diagnostics reagents. Such lack of organisational structures and resources, that is, inadequate capacity, would mean these facilities are not ready and suitable for providing internship training, for example, a qualitative study in Kenya suggested that these county hospitals are 'not organised for training purposes'.[9]

Previous studies conducted in Kenya between 2012 and 2014 highlighted poor working conditions and experiences for interns at district hospitals, however they were limited in scale (all including <10 hospitals) and did not compare hospitals at different levels and sizes.[9 11 12] Using data collected from the Kenya Harmonized Health Facility Assessment 2018 (KHFA), we address the question 'what capacity do facilities offering internship training have?' To do this, we characterise the capacity of 61 internship hospitals contrasting findings across levels of hospitals and with the minimum requirements outlined in the national guidelines for medical officer internship training,[2] with a specific focus on the 'structural' component of the

Donabedian quality of care framework (structure, process and outcome).[13]

## METHODS
### Data sources
This is a secondary data analysis using data from the 2018 KHFA. Details of the KHFA and its methodology are publicly available.[14] KHFA used the Kenya Health Master Facility List (MFL) as the sampling frame, randomly sampled 2980 out of 10 535 health facilities in Kenya, but purposely included all secondary hospitals, public primary hospitals and maternity and nursing homes. Sixty-one out of 74 internship training centres were sampled in KHFA. The survey included over 3000 questions covering five main modules on resource availability, management and finance, readiness, quality and safety of healthcare and data verification (information systems). Data were collected through facility audit, observation and validation, provider interview and record review. To supplement the KHFA data we extracted, the inpatient bed number in 2020 from the MFL[15]; the total delivery count between 2018 and 2020 from the national Kenya Health Information System (previously District Health Information Software-2 (DHIS2)); and verified key human resources for health data for six hospitals with incomplete information through direct contact with hospitals.

### Domains and indicators
We focus on the 'structure' component of the Donabedian quality of care framework to characterise hospitals. Specifically, we explore staffing, equipment and service availability and whether they fulfil the minimum requirement outlined in the Kenyan National Guidelines for Internship Training of Medical and Dental Officer Interns (2019) and the National Guidelines and Log Book for Medical Officer Interns (2019).[2] For example, the guidelines require that each internship training centre: (i) has a minimum of five medical specialists, covering paediatrics and child health, general surgery, internal medicine, OBGYN and family medicine, (ii) is fully operational on a 24-hour basis including accident and emergency, diagnostic and pharmacy services, (iii) enables interns to observe and/or perform certain procedures to fulfil competency for each specialty. Examples of the latter in surgery include a total of 36 competency procedures such as appendicectomy or repair of inguinal hernias. This requires that these services be provided by internship training centres.[2]

Two authors (ME, YZ) initially filtered out 277 indicators from the 3000 questions from KHFA to identify those most relevant to the minimum criteria defined by the national internship guidelines. Two authors (BO, FW) who are medical school faculty and who supervise medical officer interns subsequently reviewed these indicators identifying those felt to be most useful to characterising hospitals as contexts for internship training by. A final list of 166 indicators was agreed by all four authors and

**Table 1** Domain and indicator summary

| Category | Domain | Number of indicators | Example indicators | Indicator and domain conversion |
|---|---|---|---|---|
| Human resources | (a) Human resource for health | 6 | Total staff assigned, employed, seconded (including part time)—paediatricians, neonataologists | All the indicators were converted into binary responses with 1 representing 'available' and 0 representing 'unavailable'. For most indicators on availability, we defined available as 'onsite, observed, non-expired and functional'. For tests that are reported to be available offsite, or tests that are observed but expired, and for equipment reported available but not observed, or not available now, we labelled them as 'unavailable'. We also considered an indicator to be unavailable if the health facility did not answer this question, usually because its filter questions were answered no. |
| Diagnostics and supportive care | (b) Laboratory test (including four subdomains, rapid test, basic lab test, infectious diseases test, advanced lab test) | 31 | TB test availability—Xpert MTB/RIF rapid diagnostic testing for TB | |
| | (c) Oxygen and respiratory support | 10 | Please tell me if the pulse oximeter are available anywhere in the outpatient service area and are functional | |
| | (d) General equipment | 8 | ECG equipment available and functioning today | |
| Service continuity and safety | (e) 24/7 availability | 8 | Is emergency medicine specialist or general medical practitioners always available 24 hours for emergency service, either onsite in emergency unit or not onsite in emergency unit but on-call inside facility? | |
| | (f) Infectious prevention and control | 9 | Does this facility have guidelines or protocols for cleaning the facility such as the floors, counters, and beds? | |
| | (g) Quality and safety | 24 | How frequently does the quality assurance committee meet?—monthly | |
| Major specialties | (h) Surgery (including two subdomains, equipment/medicine and service) | 20 | Functioning of basic surgical equipment—ECG electrodes | |
| | (i) Internal medicine (including two subdomains, equipment/medicine and service) | 12 | Do providers in this facility diagnose and/or manage diabetic patients? | |
| | (j) Obstetrics-gynaecology (including two subdomains, equipment/medicine and service) | 11 | Parenteral administration of antibiotics (intravenously or intramuscularly) for mothers carried out | |
| | (k) Paediatrics (including two subdomains, equipment/medicine and service) | 20 | Is kangaroo mother care for premature/very low birthweight babies used in this facility? | |
| | (l) Mental health | 6 | Does this facility offer any services for mental and/or neurological conditions? | |

TB, tuberculosis.

selected for further analysis as 'signal indicators' of the minimum requirements for internship training centres. All the indicators were converted into binary responses with 1 representing 'met/available' and 0 representing 'not met/unavailable'. We grouped indicators into 12 logical domains described in brief in table 1 with further detail of all indicators available in online supplemental material 1.

### Analysis
For each domain, we calculated a score as a percentage based on the number of indicators with a score of 1 (met/available)

divided by the total number of indicators (those met/available+those not met/unavailable). We generated an overall capacity index as the unweighted mean of all 12 domain scores for each facility. We stratified the 61 included internship training centres into three categories based on hospital administrative level and inpatient bed number. Kenya has a six service-level system based on the essential package for health, whereas level 1 refers to community-level, level 2 dispensaries and clinics, level 3 health centres, level 4 primary facilities including district hospitals, level 5 secondary provincial hospitals and level 6 tertiary and national hospitals. We first grouped all internship hospitals that are level 5 and 6 together (bed number 174–1455, n=15) as historically they have been better resourced; for the internship hospitals that are level 4, we divided them based on bed number into: level 4 small hospitals (bed number 82–175, n=23) and level 4 large hospitals (bed number 176–320, n=23). Data analysis was conducted using Stata V.16 (StataCorp, Texas, USA).

### Patient and public involvement

Patients and the public were not directly involved in the design, recruitment or conducting of this study.

### RESULTS

Sixty-one out of 74 internship training centres are sampled in KHFA and are mostly public hospitals owned by the Ministry of Health (n=54). The 13/74 hospitals not sampled are either private hospitals (n=4), faith-based or mission hospitals (n=8) or a military hospital (n=1). Comparing hospital characteristics retrieved from DHIS2 and MFL between the 61 KHFA-sampled hospitals with 12 of the 13 other hospitals (the military hospital did not report to DHIS2 or MFL), the 61 KHFA hospitals are slightly larger in terms of hospital beds (median and IQR 200 (156, 276) vs 180 (106, 221)), and report conducting three times as many deliveries (median annual deliveries 3981 vs 1150) and twice as many caesarean sections (median 987 vs 537 per year). Detailed characteristics comparing the 61 KHFA internship training centres and

these 12 others are presented in online supplemental material 2.

### Overall capacity

The distribution of capacity score by domain is presented in figure 1 and table 2. Ideally, an internship centre should score 100% against the (minimum) signal indicators but the overall capacity index is 69% (95% CI 66% to 72%) for all internship training centres, and 63%, 67% and 81% for level 4 small, level 4 large and level 5 and 6 hospitals, respectively. Hospitals have moderate capacity for most of the general domains other than quality and safety. Surgery and OBGYN have higher scores regardless of hospital size, level and location, suggesting that the capacity for these two specialties is better across all facilities. Level 5 and 6 facilities have higher capacity scores in all domains when compared with level 4 hospitals.

### Human resources

The 61 internship training hospitals had a median of 14 postlicensure medical officers, 4 medical specialists, 25 clinical officers, 210 nurses and midwives, 14 medical laboratory technologists, 5 radiographers and 14 allied health professionals. Differences between small and large level 4 hospitals were small for most cadres, whereas level 5 and 6 hospitals have, on average, double the numbers of specialists, nurses and midwives (table 3). The national guideline specifically requires at least five medical specialists to act as the main clinical teachers and supervisors for medical officer interns. However, only 29 out of 61 internship training centres had at least five medical specialists assigned, employed or seconded (including part-time staff). Most hospitals had at least one surgeon (n=59/61), paediatrician (n=52/61) and an obstetrician-gynaecologist (n=56/61), whereas only 35/61 hospitals have at least an internal medicine specialist. Although not a requirement from the regulator, only 30/61 hospitals have a physician anaesthesiologist (figure 2) with anaesthesia mostly provided by clinical officers (51/61 hospitals have at least one anaesthetist clinical

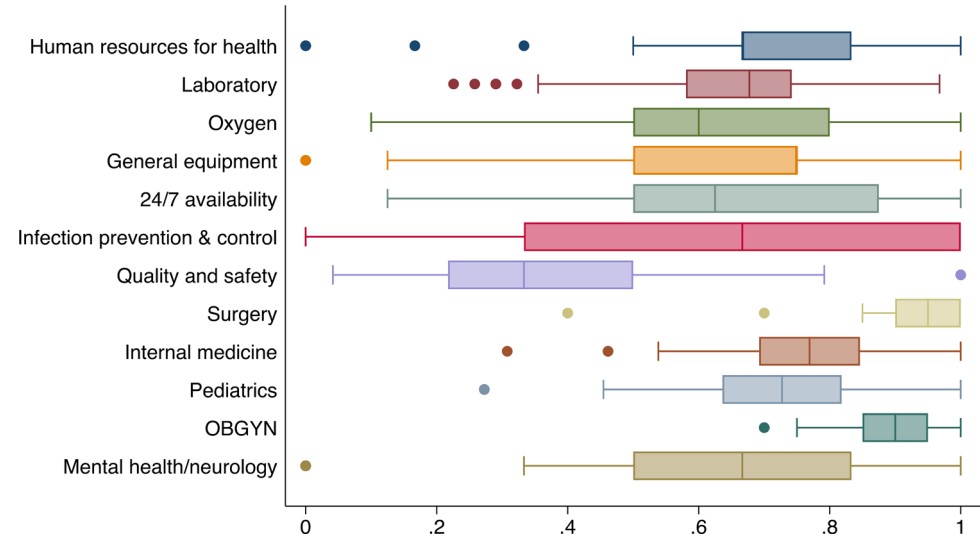

**Figure 1** Distribution of capacity score by domain.

**Table 2** Capacity scores by hospital characteristics

| | Level 4 small hospitals (n=23) | Level 4 large hospitals (n=23) | Level 5 and 6 hospitals (n=15) | Total (n=61) |
|---|---|---|---|---|
| Total capacity index | 63% | 67% | 81% | 69% |
| Human resources for health | 64% | 67% | 89% | 71% |
| Lab test | 60% | 63% | 77% | 65% |
| Oxygen and respiratory support | 64% | 57% | 77% | 64% |
| General equipment | 53% | 59% | 72% | 60% |
| 24/7 | 58% | 65% | 68% | 63% |
| Quality and safety | 35% | 38% | 50% | 40% |
| Infection prevention and control | 57% | 58% | 92% | 66% |
| Surgery | 94% | 89% | 97% | 93% |
| Internal medicine | 70% | 73% | 90% | 76% |
| Paediatrics | 67% | 70% | 81% | 72% |
| OBGYN | 89% | 91% | 91% | 90% |
| Mental health/Neurology | 44% | 73% | 83% | 64% |

The colours in the figure reflect the relative values of capacity scores, the darkest green represents the highest values in the table, while the darkest red represents the lowest values.
OBGYN, obstetrics and gynaecology.

officer). The four well-established university teritary hospitals have better human resources and all but one fulfilled all the requirements.

### Diagnostics and supportive care

As for laboratory tests, internship training centres overall have moderate availability (over 60%) for rapid tests and basic laboratory lab tests (figure 3 and online supplemental file 3), although ability to offer urine dipstick testing was present in fewer than 50% hospitals (n=26/61); 54/61 hospitals have rapid HIV tests, 16 hospitals provided onsite PCR testing for HIV but only 5 had onsite HIV viral load testing. An additional 19 hospitals relied on off-site PCR testing. Availability for advanced lab tests is poor across all hospitals, only 12/61 hospitals can read papanicolaou test results and prepare and examine any tissues or samples for cancer diagnosis onsite. Level 5 and 6 hospitals on average have better availability of

diagnostics than level 4 hospitals, especially for infectious disease and advanced lab tests.

Internship training hospitals vary when assessed for 10 oxygen and respiratory support indicators. While all hospitals were reported to have functional paediatric and neonatal oxygen, only 32 hospitals have ventilators available for adults and 13 hospitals had continuous positive airway pressure (CPAP) available for neonates (at the time of survey in 2018 pre-COVID-19). Lastly for other general equipment, around 55 hospitals reported functioning ultrasound and X-ray services. In comparison, functional ECG (27/61) and CT scan (21/61) availability was poor in most facilities.

### Service continuity and safety

The national guideline requires that internship hospitals need to be fully operational on a 24-hour basis including accident and emergency, laboratory diagnostic services and pharmacy

**Table 3** Human resources for health cadres by hospital characteristics

| | Level 4 small hospitals (n=23) | Level 4 large hospitals (n=23) | Level 5 and 6 hospitals (n=15) | Total (n=61) |
|---|---|---|---|---|
| Medical officer | 12 (6, 18) | 14 (10, 20) | 29 (21, 58) | 14 (10, 26) |
| Medical specialist | 2 (1, 7) | 4 (1, 11) | 16 (2, 26) | 4 (1, 11) |
| Clinical officer | 20 (13, 26) | 25 (15, 33) | 35 (21, 44) | 25 (15, 33) |
| Nurse and midwife | 206 (95, 248) | 150 (113, 210) | 336 (267, 534) | 210 (114, 323) |
| Medical laboratory technologist | 13 (11, 14) | 14 (11, 18) | 23 (19, 46) | 14 (12, 20) |
| Radiographer | 4 (3, 5) | 4 (2, 6) | 9 (6, 22) | 5 (3, 7) |
| Allied health professional | 11 (3, 14) | 14 (10, 17) | 29 (24, 44) | 14 (6, 24) |

Data are presented as median (IQR), allied health professional refers to physical therapist, occupational therapist, orthopaedic technician, paster technician, nutritionists and dietitians.

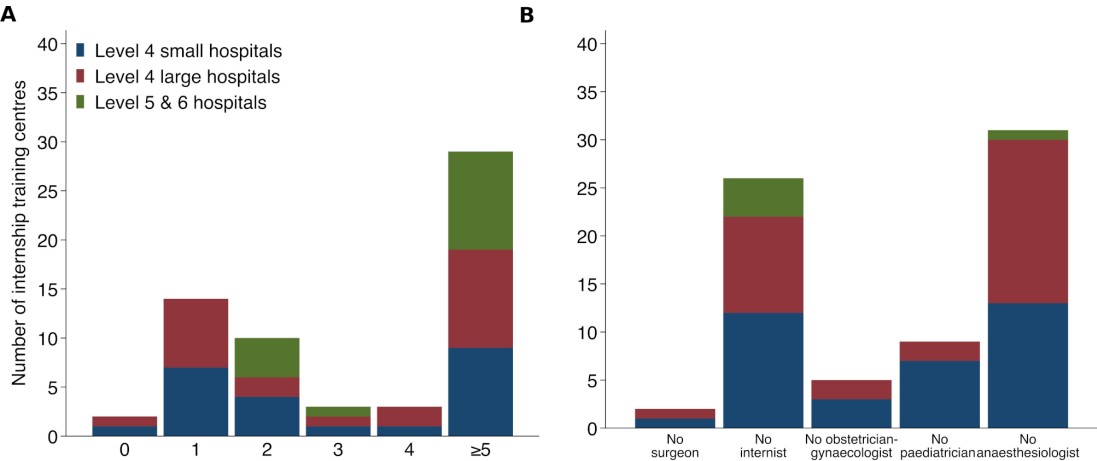

**Figure 2** Number of internship training centres with different numbers of specialists (A) and with no surgeon, internists, obstetrician-gynaecologist, paediatrician and anaesthesiologist (B). Note: We have combined different types of surgeons (general surgeons, paediatric surgeons, orthopaedic surgeons, cardiothoracic surgeon, ear, nose and throat surgeons, plastic surgeons, neurosurgeons, urological surgeons) into 'surgeon' category. Similarly, specialist physicians (internists), oncologist, neurologist, cardiologist, critical care, gastroenterologist, palliative care specialist, nephrologists, rheumatologist, medical endocrinologist into 'internist' category and paediatricians and neonatologists into 'paediatrician' category.

which was investigated under '24/7 availability'. Forty-eight out of 61 hospitals reported having general medical officers or emergency medicine specialists onsite or on-call within facility. While emergency radiology, laboratory diagnostics (other than rapid test) and pharmacy services are available 24/7 in most facilities, only 13/61 hospitals reported availability of major emergency surgery and anaesthesia 24/7, and only 16 hospitals provide 24-hour emergency services using a structured triage tool such as the WHO integrated triage tool.

The domain average score for infection prevention and control is 66%. Most internship training centres have infection prevention and control (IPC) guidelines and technical IPC committees, however, the actual procedures of cleaning beds, counters/tables, toilets were only done in around half of the hospitals on the day of KHFA survey observation. Twenty-one input/structure indicators on quality and safety are also included in the current analysis. All but one indicator (death review results recorded) performed poorly across all facilities. Most facilities do not have observed guidelines for adverse event reporting, nosocomial infection reporting, surgical adverse event reporting or postoperative infection definitions. Level 5 and 6 hospitals do not necessarily perform better than level 4 hospitals in this domain.

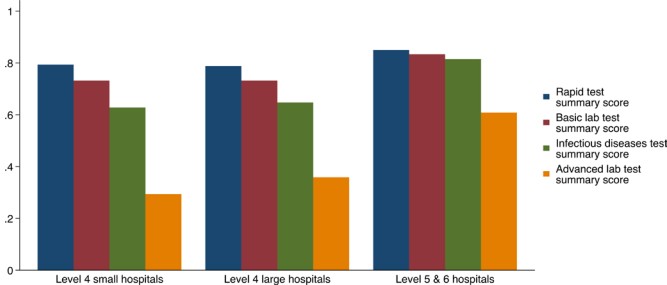

**Figure 3** Average lab test score by subdomain and hospital characteristics.

### Major specialties

For the four major specialties we used data on both specialty-specific equipment and medicines and specialty service areas (online supplemental material 3). For surgical and OBGYN scores, internship training centres in general have both functional equipment and available medicine (median 100% for surgery, 100% for OBGYN) and also good service availability for procedures relevant to the national guideline and log book (median 100% for surgery, 90% for OBGYN). The only indicator that had a lower score is cricothyroidotomy, where only 43/61 hospitals provide such service; and misoprostol tablet availability, where 44/61 hospitals have this medication available.

For internal medicine, availability of equipment and medicine is low (median 67%) with huge variation across facilities. Thirty-three out of 61 hospitals have a defibrillator, 39/61 have a renal dialysis/haemodialysis machine and 23/61 hospitals have lumbar puncture kits—for the rest of the hospitals they will need to improvise to perform lumbar punctures or cannot provide such a service. Service availability for internal medicine is good on average (median 90%), however availability for more advanced conditions, for example, colorectal cancer screening, diagnosis and/or treatment, and palliative care is limited especially for level 4 small hospitals. Similar with internal medicine, the availability of equipment for paediatrics is lower as compared with other specialties (median 63%). Only 19/61 hospitals had exchange blood transfusion capacity and similarly only 19/61 hospitals had devices for intraosseous (IO) access. The three indicators of paediatric service availability (kangaroo mother care, sepsis, antibiotics regimen and follow-up) had good availability.

Medical officer interns are also required to rotate in mental health for 8 weeks during their 1 year internship

and are required to manage acute mental health emergencies and common mental health conditions. While 52/61 hospitals offer mental health and/or neurological services, only 17 hospitals have a physician psychiatrist. Sixteen hospitals have mental health inpatient wards, other hospitals either only provide outpatient services or lack these services. Only half of level 4 small hospitals offer depression, psychosis and epilepsy diagnosis and follow-up services.

## DISCUSSION

Our analysis on the capacity of Kenya internship training hospitals suggested that in 2018 the overall staffing, equipment and service availability was inadequate when compared with the regulator's requirements. Level 4/district, smaller hospitals (23 of the 61 assessed) were more likely to have fewer human resources, equipment and specialty services, and less likely to meet quality and safety indicators. The implication is that hospitals with lower capacity may either have to improvise to offer core services or refer patients to other facilities. This could significantly influence the quality of training as interns might not be able to fully consolidate their knowledge and skills.

Over half of the internship training hospitals did not have five or more specialists as required by the regulator, the medical council, which suggests inadequate training and supervision of medical officer interns. Internal medicine specialists and anaesthesiologists were most often lacking suggesting that in many centres non-specialist but licensed general medical or specialist clinical officers fill these roles. Even in these larger Kenyan hospitals such personnel will also likely therefore be leading provision of care for severe COVID-19. Where medical officers are themselves also scarce, medical officer interns may become the only 'in-house' doctor on call and be forced to take on full responsibility for patient care potentially threatening patient safety if interns are forced to care for serious, acute medical and surgical conditions without appropriate supervision.[9 16 17] The likely challenges to provision of high quality safe care are further suggested by poor scores of all but tertiary hospitals against organisational indicators of quality and safety practice. In fact guidelines or instruction for quality and safety were rarely present and there seemed very limited evidence that quality and safety indicators are monitored, reviewed, reported and acted on. These findings are consistent with reports highlighting challenges with patient safety in LMIC hospitals.[18]

Internship hospitals appear most able to provide appropriate essential resources and capacities in the major disciplines of surgery and OBGYN. Challenges were especially noted in the disciplines of internal medicine and paediatrics and neonatology. Many facilities lack ECG machines, CT scans, defibrillators, equipment used for peritoneal or haemodialysis, and lumbar puncture kits. More specific to neonatal and child health were lack of capacity in ability to provide exchange transfusion, neonatal CPAP and IO access. Capacity to offer mental health and neurological care was especially low most particularly in level 4 small hospitals, likely due to these only being listed as internship requirements in 2020, and that Kenya has a severe shortage of psychiatrists and neurologists as well as psychiatric nurses.[19] For facilities without capacity in the major disciplines now including mental health and neurology services, it is likely that hospitals are forced to refer patients elsewhere or improvise approaches to care in order to serve patients. Interns' training experience is therefore likely to be very varied and more attention may therefore need to be paid to planning how interns can be provided with adequate training in all specialties if this is a key long-term aim.

Kenya has dramatically increased its medical training volume hoping to produce 9000 new graduate medical physicians by 2030 to narrow its staffing gap.[6] More hospitals especially at county-level (district hospitals) are accredited as internship training centres. These hospitals are usually smaller in size and more likely to be distant from the well-established university tertiary hospitals in major cities that scored much higher across domains in our analysis. Rural rotations and residencies have been recommended by WHO to increase health worker retention.[20] Students or interns in these settings may also be more likely to learn hands-on clinical procedures and be actively engaged in patient care[21] as opposed to tertiary hospitals where the presence of more specialists and general medical officers may result in interns undertaking more administrative work. However, the prerequisite of good internship training is that the hospitals are adequately staffed and equipped and ready to deliver teaching and training. Our data suggested that this is not always the case and are consistent with previous qualitative research suggesting that district hospitals provided limited learning opportunities and supervision with limited suitability as internship sites.[9]

The potential consequences of poor training and supervision during internship are therefore worth consideration. As well as failing to consolidate knowledge and skills in major specialties interns may also develop burnout and stress-related psychological problems more rapidly.[9] These stresses and being forced to take significant clinical responsibility in poorly resourced hospitals may push medical officer interns to leave the profession or the public sector which is often most resource constrained as soon as they are licensed/registered.[22] This will likely worsen the internal and external brain drain undermining universal health coverage and equity and limiting any benefits from public investment in medical education.[22] Inadequate support and supervision also threatens patient safety by creating the conditions for significant medical errors to occur. Even in settings without a 'blame culture' such errors can have profound effects on the health workers as well as the patients further exacerbating workers' psychological distress.[23 24] We must also remember that today's medical officer interns will become tomorrow's general

medical officers who are often then responsible for the training of the next cohorts of interns and other health worker cadres.[25] Therefore, poor internship training for medical officers continuously compromises the quality of training and skills that are passed on and this may be a particular challenge to patient safety and healthcare quality that rely so heavily on effectively functioning teams.[26]

Our analyses have policy implications for Kenya policymakers. The regulator requires internship hospitals to maintain the minimum requirement of staff and be able to offer a core set of quality services with self-report against such requirements at least once a quarter. Our data would suggest many internship hospitals in Kenya might not continuously achieve these minimum requirements. Should this prevent interns from being licensed or should interns be reallocated to centres meeting regulatory requirements? While more stringent and regular audit and re-accreditation of internship training centres might be conducted by the regulatory council to ensure that only the hospitals that meet the minimum requirements are allowed to receive and train interns, adequate mitigation measures need to be in place so that interns themselves are not disadvantaged further. For example, by rotating interns between different level hospitals with different level of resource availability. More widely, our findings point to the need to carefully consider the potential consequences of rapidly expanding medical training and of appropriate planning and financing for new internship centres, especially in rural areas.

Several limitations should be noted for the current analysis. To start with, our analysis was only limited to the 61 internship training hospitals sampled by KHFA, mostly public hospitals. A total of 13 hospitals were not sampled in KHFA and are either private hospitals or mission hospitals, therefore comparison by hospital ownership was not feasible. Second, aside from the KHFA's own limitation on data missing, we noted data inaccuracy and inconsistency in KHFA data we retrieved. We made efforts to clean these data especially on human resources through correspondence input, although we were unable to validate all the number from the KHFA survey. Third, to assess the capacity of internship training hospitals we selected 166 signal indicators from the 3000 questions from KHFA survey. Our criteria were to ensure that the indicators selected are the minimum requirement and should be achievable (ie, should be '1' for all indicators). Our selection was guided by the national guidelines and agreed by four authors, three of whom have experience supervising interns. However, we acknowledge that this process is somewhat subjective. Fourth, we only focused on the structural and organisational features of internship training hospitals. We did not include indicators on process or outcome indicators, for example, training or patient outcomes due to limited data availability. Ideally, these should be considered while evaluating internship training hospitals. Last, the KHFA data were collected in a snapshot in 2018 prior to COVID-19. The appointment

and posting of specialists may fluctuate from time to time but COVID-19 specifically could have led to the government investing in hospital infrastructure, equipment and emergency hiring to improve their response capacity to COVID-19. Therefore, it is possible that the capacity of Kenyan internship training centres has improved since 2018.[27] Despite these limitations, our data do suggest important shortcomings in internship training centres in Kenya. We suggest other LMICs that are rapidly expanding their medical training should also evaluate their internship training sites to explore the generalisability of our findings.

## CONCLUSION

We assessed the capacity of 61 Kenyan internship training hospitals using data from KHFA 2018. Our results highlighted major gaps in staffing, equipment and service availability. More specifically, there are weaknesses in areas regarding organisational arrangements that support quality and safety. Smaller hospitals are more likely to have a lower capacity index, and should be re-accredited more stringently and regularly and also be provided with adequate mitigation measures so they can provide appropriate and well-resourced training for medical interns allowing them to become fully competent medical doctors.

**Author affiliations**
[1]Oxford Centre for Global Health Research, Nuffield Department of Medicine, University of Oxford, Oxford, UK
[2]Department of Paediatrics and Child Health, University of Nairobi, Nairobi, Kenya
[3]Kenya Paediatric Research Consortium (KEPRECON), Nairob, Kenya
[4]Division of Monitoring & Evaluation, Ministry of Health, Nairobi, Kenya
[5]Nuffield Department of Primary Care Health Sciences, University of Oxford, Oxford, UK
[6]Wellcome Trust Research Program, Kenya Medical Research Institute, Nairobi, Kenya
[7]KEMRI-Wellcome Trust Research Programme Nairobi, Nairobi, Kenya

**Contributors** YZ and ME conceived of the analysis. YZ, BO, FW and ME contributed to data analysis. YZ wrote the first draft of the manuscript. BO, FW, HK, DG, CN and ME provided critical feedback on the first draft of the manuscript. All authors read and approved the final manuscript. YZ is the gurantor and accepts full responsibility for the work and/or the conduct of the study, had access to the data, and controlled the decision to publish.

**Funding** YZ is supported by the University of Oxford Clarendon Fund Scholarship. ME is supported by a Wellcome Trust Senior Research Fellowship (#207522). HK is employed by the Kenya Ministry of Health.

**Competing interests** None declared.

**Patient and public involvement** Patients and/or the public were not involved in the design, or conduct, or reporting, or dissemination plans of this research.

**Patient consent for publication** Not applicable.

**Ethics approval** The Kenya Ministry of Health, Division of Monitoring and Evaluation, sought approval for, planned and executed the 2018 Kenya Harmonized Health Facility Assessment. Written informed consent was sought by in-charge or acting in-charge of all the facilities surveyed. Permission was then granted to use these data for the secondary analysis reported here with the purpose of improving quality of health service provision in Kenya.

**Provenance and peer review** Not commissioned; externally peer reviewed.

**Data availability statement** Data may be obtained from a third party and are not publicly available. The data that support the findings of this study are available from Kenya Ministry of Health but restrictions apply to the availability of these data, which were used under license for the current study, and so are not publicly available. All requests for further use of these data can be made through and with the permission of the Kenya Ministry of Health.

**ORCID iDs**
Yingxi Zhao http://orcid.org/0000-0002-4937-4703
Catia Nicodemo http://orcid.org/0000-0001-5490-9576
Mike English http://orcid.org/0000-0002-7427-0826

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
