## [Reviewer comments · BMJ Open]

ARTICLE DETAILS

TITLE (PROVISIONAL)	Characterising Kenyan hospitals' suitability for medical officer internship training: secondary data analysis of a cross-sectional study
AUTHORS	Zhao, Yingxi; Osano, Boniface; Were, Fred; Kiarie, Helen; Nicodemo, Catia; Gathara, David; English, Mike

VERSION 1 – REVIEW

REVIEWER	Ahmed, Mushtaq The Aga Khan University
REVIEW RETURNED	21-Dec-2021

GENERAL COMMENTS	Your paper provides convincing quantitative evidence of the inadequacy of district hospitals in Kenya for medical internship training; it nicely complements an earlier qualitative study which came to the same conclusion (1). You draw attention to policy implications for stricter implementation of accreditation standards to ensure suitability of hospitals for internship training and warn against an unduly rapid expansion of medical schools that could leave medical graduates stranded without suitable internship training. Arguably a more pressing reason for enhancing capacity and quality of service in district hospitals, not only in Kenya but in Low- and Middle- Income Countries in general, is to decentralize health services to meet the requirements of a growing burden of Non-Communicable Disease and Global Surgery (2,3). The district hospital would serve as the hub of a health system that incorporates Primary Health Care to enhance access to care especially in disadvantaged areas. Likewise, diversification of clinical experience for medical and other health professional students to district health services would ensure that the context is right for developing competencies leading to retention of graduates who are fit for purpose (4,5). You rightly point out that it is time to pay attention to the need for congruent development of health and education systems to achieve Universal Health Coverage. References: 1. Muthaura P, Khamis T, Ahmed M, et al. Perceptions of the preparedness of medical graduates for internship responsibilities in district hospitals in Kenya: a qualitative study. BMC Medical Education 2015; 15: 178 2. Bukhman G, Mocumbi AO, Atun R, et al. The Lancet NCDI Poverty Commission: bridging a gap in universal health coverage for the poorest billion. Lancet. 2020; 396:91-1044. 3. Meara JG, Leather AJ, Hagander L, et al. Global Surgery 2030: evidence and solutions for achieving health, welfare, and economic development. Lancet. 2015; 386:569-624.
---

	4. Talib Z, van Schalkwyk S, Couper I, et al. M. Medical education in decentralized settings: How medical students contribute to health care in 10 sub-Saharan African countries. Acad Med. 2017; 92:1723. 5. Hiatt RA, Engmann NJ, Ahmed M, et al. Population health science: A core element of health science education in Sub-Saharan Africa. Acad Med. 2017; 92:462. Post script: You may wish to review the range of bed numbers for small v. large district hospitals in your study; the number 176 is included in both sets of ranges.
--	--

REVIEWER	Naidoo, Kimesh King Edward VIII Hospital, Nelson R Mandela School of Medicine University of KwaZulu-Natal, Paediatrics and Child health
REVIEW RETURNED	06-Jan-2022

GENERAL COMMENTS	This is an important study that attempts to assess capacity for medical internship training in a resource constrained context. The authors use secondary data to review internship accreditation. The findings are important . A major issue is whether interns rotate within different level hospitals .If this is the case they will have exposure to differing resourced hospitals ,if not this could be a suggestion to improve exposure. Line by line pointers have been made - see attached reviewer table Line correction suggested Comment Suggestion Title A bit too long Characterising Kenyan hospitals' suitability for medical officer internship training Abstract Objective Line 35 First sentence confusing Use of the term quality – does this refer to intern training or patient care or both Remove quality and focus on readiness and capacity only Rewrite sentence setting and population Rewrite Remove 'we used secondary data....as its repeated from statement in the design ? Sixty one Kenyan hospitals that train medical interns Results Line 48 'Assigned, employed, seconded Use one term? available for internship training Line 52 Change level 5&6 and Level 4 Abstract readers may be unfamiliar with these levels Replace sentence to indicate referral hospitals have higher capacity than smaller, entry level hospitals. Conclusion Line 55 Change level 4 Use different term? smaller ,entry level hospitals Background Line 87 Need to expand a bit more Do Kenyan interns work in hospital complexes (level 4 and level 5/6 hospitals jointly) so that they rotate as this could be one way they get exposed to differing levels of care If they are not rotating this could be a recommendation of this study (especially where differing levels of hospitals have such different capacity) Background
--

	Line 91 Need to be more specific What's the difference between the guidelines and the logbook – need to indicate this Are there compulsory procedure lists etc. that need completion Line 91-92 Need to clarify if once licenced can these medical officers work independently in private practices as well Line 99 Sentence starting 'Ensuring internshipis too long Separate into two sentences 1. Deals with training and support 2. The situation in LMIC hospitals where interns are default frontline staff due to inadequate staffing numbers Line 105 The word 'expanding ' Could mean many things – be more specific Increasing medical student enrolment to produce larger numbers of newly qualified doctors Line 108 Remove correspondingly Line 112 You have mentioned levels without indicating what these levels are The explanation of the levels provided in lines 179-182 can be brought forward to this point. Line 122 Need to define what you mean by ' readiness and capacity ' in this study first Are they different and how are they different Line 126 Need a reference for the Donabedian quality of care = also need to introduce briefly- what does Donabedian quality framework entail ? structure ,process ,outcomes Methods Line 138 Can you define what readiness means in this context Patient related or training related Lines 179 – 182 As previous comment take to the introduction section Line 189 Patient and public involvement Consider adding to the ethics section Results Table 1 In the column example indicators – needs simplifying or removing most reader swill not have access to the KHFA so either remove question numbers Indicator domain and conversion Need to clarify ' for most indicators " – do you have a number NB the binary conversion should be listed as a limitation unless you verified all results that were " unavailable " - Some institutions who did not answer a question will be scored less Discussion Lines 338 -341 You as an make two points her regarding mental health being listed internship requirement and Kenya's shortage of psychiatrists You could postulate the link as the primary reason capacity indices are so low for this aspect Line 344 Nee to consider interns rotating between different levels of hospitals as a means to provide exposure Line 361 Was there limited supervision as well – need to clarify Was the limited supervision compromising intern learning Line 382 Consider changing the word 'clear " Either remove or
--	---

	change Line 396 Need to add the limitation that binary coding may have compromised the results (stated earlier) Line 412 Remove ‘ last but not least” Conclusion – needs to be rewritten Line 424 Second sentence in conclusion too long and not easily understood Separate sentences and concepts you trying to get across Line 427 What do you mean by greater support Ethics Line 468 Did this secondary analysis get regulatory approval from an academic institution Reference Review references 2,4 ,7 and 8 (need dates of citation in some and more detail)
--	--

VERSION 1 – AUTHOR RESPONSE

Reviewer: 1

Dr. Mushtaq Ahmed, The Aga Khan University

Comments to the Author:

1. Your paper provides convincing quantitative evidence of the inadequacy of district hospitals in Kenya for medical internship training; it nicely complements an earlier qualitative study which came to the same conclusion (1). You draw attention to policy implications for stricter implementation of accreditation standards to ensure suitability of hospitals for internship training and warn against an unduly rapid expansion of medical schools that could leave medical graduates stranded without suitable internship training.

Arguably a more pressing reason for enhancing capacity and quality of service in district hospitals, not only in Kenya but in Low- and Middle- Income Countries in general, is to decentralize health services to meet the requirements of a growing burden of Non-Communicable Disease and Global Surgery (2,3). The district hospital would serve as the hub of a health system that incorporates Primary Health Care to enhance access to care especially in disadvantaged areas. Likewise, diversification of clinical experience for medical and other health professional students to district health services would ensure that the context is right for developing competencies leading to retention of graduates who are fit for purpose (4,5). You rightly point out that it is time to pay attention to the need for congruent development of health and education systems to achieve Universal Health Coverage.

References:

1. Muthaura P, Khamis T, Ahmed M, et al. Perceptions of the preparedness of medical graduates for internship responsibilities in district hospitals in Kenya: a qualitative study. *BMC Medical Education* 2015; 15: 178
2. Bukhman G, Mocumbi AO, Atun R, et al. The Lancet NCDI Poverty Commission: bridging a gap in universal health coverage for the poorest billion. *Lancet*. 2020; 396:91-1044 PubMed .
3. Meara JG, Leather AJ, Hagander L, et al. Global Surgery 2030: evidence and solutions for achieving health, welfare, and economic development. *Lancet*. 2015; 386:569-624 PubMed .
4. Talib Z, van Schalkwyk S, Couper I, et al. M. Medical education in decentralized settings: How medical students contribute to health care in 10 sub-Saharan African countries. *Acad Med*. 2017; 92:1723.

5. Hiatt RA, Engmann NJ, Ahmed M, et al. Population health science: A core element of health science education in Sub-Saharan Africa. Acad Med. 2017; 92:462.

Thanks for the comment. We do agree that this complements the earlier qualitative study by Muthaura et al. and believe enhancing readiness and capacity of district hospitals is a pressing issue not only for Kenya and also other LMICs.

1. Post script: You may wish to review the range of bed numbers for small v. large district hospitals in your study; the number 176 is included in both sets of ranges.

Thanks for the comment. We have revised the range of bed number into 82-175 and 176-320 (see line 190). Sorry for the confusion.

Reviewer: 2

Dr. Kimesh Naidoo, King Edward VIII Hospital, Nelson R Mandela School of Medicine University of KwaZulu-Natal

Comments to the Author:

1. This is an important study that attempts to assess capacity for medical internship training in a resource constrained context. The authors use secondary data to review internship accreditation. The findings are important. A major issue is whether interns rotate within different level hospitals. If this is the case they will have exposure to differing resourced hospitals, if not this could be a suggestion to improve exposure. Line by line pointers have been made - see attached reviewer table

Thanks for the comment. For internship training in Kenya, interns are usually posted to one single facility for a total of 12 months without rotating within different level hospitals. We do agree that rotating between different level hospitals could be one suggestion to improve exposure and have added this in the policy recommendation section in line 391-393.

“While more stringent and regular audit and re-accreditation of internship training centres might be conducted by the regulatory council to ensure that only the hospitals that meet the minimum requirements are allowed to receive and train interns, adequate mitigation measures need to be in place so that interns themselves are not disadvantaged further. For example, by rotating interns between different level hospitals with different level of resource availability.”

1. Line correction suggested Comment Suggestion

- 1) Title A bit too long Characterising Kenyan hospitals' suitability for medical officer internship training

Thanks for the comment. We have revised the title accordingly.

- 2) Abstract
Objective
Line 35 First sentence confusing
Use of the term quality – does this refer to intern training or patient care or both Remove quality and focus on readiness and capacity only

Thanks for the comment. We have removed the term quality.

- 3) Rewrite sentence
setting and population Rewrite
Remove 'we used secondary data....as its repeated from statement in the design
? Kenyan hospitals that train medical interns

Thanks for the comment. We have revised the sentence.

- 4) Results Line 48
'Assigned, employed, seconded Use one term? available for internship
training

Thanks for the comment. These terms are the original terms from the KHFA survey, we believe stating this is important as some consultants are not fully employed by the internship hospitals (e.g. seconded from medical schools) but still provide clinical work to some extent in a hospital.

- 5) Line 52
Change level 5&6 and Level 4
Abstract readers may be unfamiliar with these levels Replace sentence to indicate
referral hospitals have higher capacity than smaller, entry level hospitals.

Thanks for the comment. We have added "(provincial and national hospitals)" after "level 5 and 6", and "(equivalent to district hospitals)" after "level 4" to further clarify.

- 6) Conclusion Line 55
Change level 4 Use different term? smaller ,entry level hospitals

Thanks for the comment. Per previous comment, we have changed the wording to clarify what level 4 hospitals mean.

- 7) Background
Line 87 Need to expand a bit more Do Kenyan interns work in hospital
complexes (level 4 and level 5/6 hospitals jointly) so that they rotate as this could
be one way they get exposed to differing levels of care
If they are not rotating this could be a recommendation of this study (especially
where differing levels of hospitals have such different capacity)

Thanks for the comment. For internship training in Kenya, interns are usually posted to one single facility for a total of 12 months without rotating within different level hospitals. We do agree that rotating between different level hospitals could be one suggestion to improve exposure. Accordingly we have added the two sentences below to the background section and the policy recommendation section.

"The Kenyan medical internship included supervised rotation in 4 major departments (surgery, internal medicine, paediatrics and child health, and obstetrics and gynaecology [OBGYN]) in one Internship training centre but since 2020 mental health and community health practice became added requirements."

"While more stringent and regular audit and re-accreditation of internship training centres might be conducted by the regulatory council to ensure that only the hospitals that meet the minimum requirements are allowed to receive and train interns, adequate mitigation measures need to be in place so that interns themselves are not disadvantaged further. For example, by rotating interns between different level hospitals with different level of resource availability."

- 8) Line 91 Need to be more specific
What's the difference between the guidelines and the logbook – need to indicate this Are there compulsory procedure lists etc. that need completion

Thanks for the comment. The guidelines and logbooks both suggested the compulsory procedures that should be carried out either independently or under supervision by interns. The difference is that the latter has more detail and “log tables” for supervisors to check. We have revised the sentence below.

“The competencies medical interns are expected to develop e.g. compulsory procedures that should be carried out either independently or under supervision, are outlined in national guidelines and in greater detail in interns’ personal log books”

- 9) Line 91-92 Need to clarify if once licenced can these medical officers work independently in private practices as well

Thanks for the comment. Once licensed MOs can work unsupervised in all different types of facilities including in private practice. We have updated the sentence accordingly.

- 10) Line 99 Sentence starting ‘Ensuring internship ...is too long Separate into two sentences
1. Deals with training and support
 2. The situation in LMIC hospitals where interns are default frontline staff due to inadequate staffing numbers

Thanks for the comment. We have divided the sentence into two.

- 11) Line 105 The word ‘expanding ‘
Could mean many things – be more specific Increasing medical student enrolment to produce larger numbers of newly qualified doctors

Thanks for the comment, we have changed the word to increase, as we are referring to the number of medical schools being increased.

- 12) Line 108 Remove correspondingly

Thanks for the comment, we have removed the word.

- 13) Line 112 You have mentioned levels without indicating what these levels are The explanation of the levels provided in lines 179-182 can be brought forward to this point.

Thanks for the comment, we have added a few words after the level to indicate that level 4 hospitals are primary facilities including district hospitals and level 5 and 6 are equivalent to provincial and national hospitals.

- 14) Line 122 Need to define what you mean by ‘readiness and capacity ‘ in this study first Are they different and how are they different

Thank you for this important feedback. In this study we primarily focus on the “capacity”, i.e. the organizational structures and resources of internship hospitals. This would decide on whether hospitals are ready and suitable or not to provide internship trainings. To avoid confusion, we have changed all “readiness and capacity” into “capacity” and have added this sentence in the introduction paragraph on line 119-122.

“Such lack of organizational structures and resources, i.e. inadequate capacity, would mean these facilities are not ready and suitable for providing internship training”

- 15) Line 126 Need a reference for the Donabedian quality of care = also need to introduce briefly- what does Donabedian quality framework entail
? structure ,process ,outcomes

Thanks for the comment, we have added the reference and explained the three components of the Donabedian model.

- 16) Methods
Line 138

Can you define what readiness means in this context Patient related or training related

Thanks for the comment, “readiness” here is the direct quote from the KHFA report, which refers to the service readiness, e.g. basic amenities, equipment, standard precaution for infection prevention, diagnostic capacity, essential medicine. We have kept this word here but deleted all the other instances of use of “readiness” in the paper.

- 17) Lines 179 – 182 As previous comment take to the introduction section

Thanks for the comment, we have kept this here but also have provided additional information in the introduction section.

- 18) Line 189 Patient and public involvement Consider adding to the ethics section

Thanks for the comment, this section is required by BMJ Open to be added under the method section thus we have left it here.

- 19) Results

Table 1 In the column example indicators – needs simplifying or removing most reader swill not have access to the KHFA so either remove question numbers

Thanks for the comment, we have removed the question number.

- 20) Indicator domain and conversion Need to clarify ‘ for most indicators “ – do you have a number

Thanks for the comment. We have provided a more detailed conversion and definition of “available/unavailable” in appendix 1. Different indicators have different answer (e.g. some questions ask about equipment available and functioning, some questions ask about certain documentations being observed).

- 21) NB the binary conversion should be listed as a limitation unless you verified all results that were “ unavailable “

- Some institutions who did not answer a question will be scored less

Thanks for the comment. We have provided a more detailed conversion and definition of “available/unavailable” in appendix 1. Whereas some institutions did not answer a question, it’s usually because they answered no to a previous filter question – e.g. if the facility do not offer “any mental/neurological services) then they would not answer questions on “depression diagnosis and follow-up available” or “psychosis diagnosis and follow-up available”, in which case we deemed for those two specific services it will also be unavailable.

We have mentioned in the discussion section that we have noted data inaccuracy and inconsistency in KHFA data we retrieved. We made efforts to clean these data especially on human resources through correspondence input though we were

unable to validate all the number from the KHFA survey

22) Discussion

Lines 338 -341 You as an make two points her regarding mental health being listed internship requirement and Kenya's shortage of psychiatrists You could postulate the link as the primary reason capacity indices are so low for this aspect

Thanks for the comment, we have rephrased the sentence see below.

“Capacity to offer mental health and neurological care was especially low most particularly in Level 4 small hospitals, likely due to these only being listed as internship requirements in 2020, and that Kenya has a severe shortage of psychiatrists and neurologists as well as psychiatric nurses (18).”

23) Line 344 Nee to consider interns rotating between different levels of hospitals as a means to provide exposure

Thanks for the comment, we have incorporated your suggestion in the policy recommendation section.

24) Line 361 Was there limited supervision as well – need to clarify
Was the limited supervision compromising intern learning

Thanks for the comment, we have added limited supervision to this sentence

25) Line 382 Consider changing the word ‘clear “ Either remove or change

Thanks for the comment, we have changed the word into “policy implications”

26) Line 396 Need to add the limitation that binary coding may have compromised the results (stated earlier)

Thanks for the comment. We have explained the rationale for binary coding in the previous point.

27) Line 412 Remove ‘ last but not least”

Thanks for the comment, it's removed.

28) Conclusion – needs to be rewritten

Line 424 Second sentence in conclusion too long and not easily understood Separate sentences and concepts you trying to get across

Thanks for the comment. We have split the sentence into two.

29) Line 427 What do you mean by greater support

Thanks for the comment. We have removed the term support into the following sentence.

“Smaller hospitals are more likely to have a lower readiness and capacity index, and should be re-accredited more stringently and regularly and also be provided with adequate mitigation measures so they can provide appropriate and well-resourced training”

30) Ethics

Line 468 Did this secondary analysis get regulatory approval from an academic institution

Thanks for the comment, we received permission from the Kenyan Ministry of Health Division of Monitoring and Evaluation for secondary analyses of their survey data and one of our co-authors is from the Ministry of Health.

31) Reference

Review references 2,4 ,7 and 8 (need dates of citation in some and more detail)

Thanks for the comment. We have added dates for reference 2 and 7, revised reference 4's formatting and added "final draft" for reference 8 as it's an internal policy document not yet publicly available.

VERSION 2 – REVIEW

REVIEWER	Naidoo, Kimesh King Edward VIII Hospital, Nelson R Mandela School of Medicine University of KwaZulu-Natal, Paediatrics and Child health
REVIEW RETURNED	19-Feb-2022
GENERAL COMMENTS	The authors have addressed all my major concerns and provided adequate explanations .